
# One for "All": a unified model for fine-grained sentiment analysis under three tasks

Heng-yang Lu[1,2], Jun Yang[3], Cong Hu[1] and Wei Fang[1]

[1] Jiangsu Provincial Engineering Laboratory of Pattern Recognition and Computational Intelligence, Jiangnan University, Wuxi, China
[2] State Key Laboratory for Novel Software Technology, Nanjing University, Nanjing, China
[3] Marcpoint Co., Ltd., Shanghai, China

## ABSTRACT

**Background**. Fine-grained sentiment analysis is used to interpret consumers' sentiments, from their written comments, towards specific entities on specific aspects. Previous researchers have introduced three main tasks in this field (ABSA, TABSA, MEABSA), covering all kinds of social media data (*e.g.,* review specific, questions and answers, and community-based). In this paper, we identify and address two common challenges encountered in these three tasks, including the low-resource problem and the sentiment polarity bias.

**Methods**. We propose a unified model called PEA by integrating data augmentation methodology with the pre-trained language model, which is suitable for all the ABSA, TABSA and MEABSA tasks. Two data augmentation methods, which are entity replacement and dual noise injection, are introduced to solve both challenges at the same time. An ensemble method is also introduced to incorporate the results of the basic RNN-based and BERT-based models.

**Results**. PEA shows significant improvements on all three fine-grained sentiment analysis tasks when compared with state-of-the-art models. It also achieves comparable results with what the baseline models obtain while using only 20% of their training data, which demonstrates its extraordinary performance under extreme low-resource conditions.

## INTRODUCTION

Consumers worldwide have posted trillions of text comments on online shopping sites and social platforms to express their opinions. The efficiency of how modern merchandisers drive insights from those opinions would be the key to their success in the data-driven era. Sentiment analysis is such a solution for businesses to understand consumers' opinions effectively. Traditional coarse-grained sentiment analysis aims to identify the sentiment polarity of the given sentence. Different from that, fine-grained sentiment analysis is managed to match sentiments with corresponding entities and aspects in the given sentence. For example, given the comment "I've used MacBookPro, it's convenient."

Corresponding author
Jun Yang, jun.yang@marcpoint.com

Coarse-grained sentiment analysis describes the whole sentence a positive sentiment. Fine-grained sentiment analysis describes a positive sentiment towards MacBookPro (entity) on its convenience level (aspect), which is a provided (sentence, aspect, entity) pair. Previous researchers have introduced three tasks on fine-grained sentiment analysis towards entities and aspects (definitions and two examples are illustrated in Table 1):

1. Aspect-Based Sentiment Analysis (**ABSA**),
2. Targeted Aspect-Based Sentiment Analysis (**TABSA**),
3. Multi-Entity Aspect-Based Sentiment Analysis (**MEABSA**).

ABSA was primarily based on the review-specific data acquired from E-commerce or life service websites (*e.g.*, Amazon, Yelp) where there is only one or even no entity mentioned in the data. Although performing well on consumer reviews, models designed for ABSA have limited performance on posts coming from social platforms (*e.g.*, Twitter, Reddit) where there are multiple entities and aspects mentioned. For example, a software engineer on Twitter wrote "I've used MacBookPro, it's convenient. But now I switched to ThinkPad because it's just as convenient and has a better price." There are two entities introduced: MacBookPro and ThinkPad. For each of the entities, sentiments on the convenience level (aspect 1) are the same while sentiments on the price (aspect 2) are different. TABSA was proposed by (*Saeidi et al., 2016*) to handle such multi-entity and multi-aspect cases. This task was based on the *SentiHood* dataset acquired from the question answering platform, which involves two entities of the same kind (*e.g.*, tourist attractions) and 15 aspects. However, in reality, not only do consumers compare entities of that same kind but also should they talk about multi-kind entities. *Yang et al. (2018)* proposed MEABSA with the *BabyCare* dataset acquired from a community-based platform. It involves hundreds of multi-kind entities (*e.g.*, powdered milk, diapers, and infant medicines) and hundreds of aspects. The increase in the number of entities and aspects makes MEABSA the most challenging task among the three.

Most previous works are designed for only one of the tasks, it is more practical to design a unified model, which is available for all three tasks. What's more, the Recurrent Neural Network (RNN)-based models (*Yang et al., 2018*; *Yang et al., 2019*; *Xu et al., 2020*) and BERT-based models (*Sun, Huang & Qiu, 2019*) are two kinds of recently proposed basic models for fine-grained sentiment analysis, which have shown effectiveness. The RNN-based models have the advantages of considering the global sequence, and the BERT-based models are good at considering local attention. It is promising to improve the predictions of sentiments by making use of both advantages.

Additionally, there are two main challenges encountered in the ABSA, TABSA, and MEABSA tasks. The first challenge is the **low-resource problem**, also known as the insufficient data problem. This is often caused by the large time and money required by manual annotation. The low-resource problem is even more prevalent in sentiment prediction towards entities and aspects due to the increasing complexity of data annotation: for example, if there are three entities and two aspects mentioned in the text, one needs to annotate 6 (3*2) instances for each of the entity aspect combinations. This explains the fact that 59% of the entity aspect combinations are annotated five times or less in the *BabyCare* dataset. The second challenge is the **polarity bias problem**. It reduces task

**Table 1** The comparison between three tasks of sentiment prediction towards entities and aspects.

| | | ABSA | TABSA | MEABSA |
|---|---|---|---|---|
| **Definition** | **Given** | text and aspects mentioned | text, entity mentioned, and all kinds of aspects | text, entity mentioned, and aspect mentioned |
| | **Goal** | predict sentiment towards mentioned aspects | predict sentiment towards the combination of mentioned entity and all kinds of aspects | predict sentiment towards mentioned entity aspect combination |
| **Example1** | **Input** | <**context**>I've used MacBookPro, it's convenient. </**context**><**aspect** from ="27″to="37″>convenience level</**aspect**> | <**context**>I've used MacBookPro, it's convenient. </**context**><**entity** from="10″to="20″>MacBookPro </**entity**><**aspectlist**>price, convenience level, battery, …</**aspectlist**> | <**context**>I've used MacBookPro, it's convenient. </**context**><**entity** from="10″to="20″>MacBookPro </**entity**><**aspect**>from="27″to="37″> convenience level</**aspect**> |
| | **Output** | (convenience level, positive) | (MacBookPro, price, none) (MacBookPro, convenience level, positive) (MacBookPro, battery, none) … | (MacBookPro, convenience level, positive) |
| **Example2** | **Input** | <**context**>The battery of ThinkPad is very long. </**context**><**aspect** from ="4″to="11″>battery</**aspect**> | <**context**>The battery of ThinkPad is very long. </**context**><**entity** from="15″to="23″>ThinkPad </**entity**><**aspectlist**>price, convenience level, battery, …</**aspectlist**> | <**context**>The battery of ThinkPad is very long. </**context**><**entity** from="15″to="23″>ThinkPad </**entity**><**aspect**>from="4″to="11″ >battery</**aspect**> |
| | **Output** | (battry, positive) | (ThinkPad, price, none) (ThinkPad, convenience level, none) (ThinkPad, battery, positive) … | (ThinkPad, battery, positive) |

performance when entities' sentiment polarity distribution is not uniform in the training set. For example, if an entity is mostly labeled positive in the training set, it will be more likely to be predicted positive regardless of the context. This problem is mainly caused by the inconsistent polarity distributions between the training set and test set from the perspective of entities.

This paper aims to propose a unified model for fine-grained sentiment analysis, which is available for ABSA, TABSA and MEABSA tasks. The main contributions of this paper include:

- To the best of our knowledge, this is the first work unifying the ABSA, TABSA, and MEABSA tasks together, providing an all-in-one solution to fine-grained sentiment analysis.
- We propose a unified model, which combines both advantages of RNN-based models and BERT-based models with ensemble methods. This model achieves outstanding performance in all the ABSA, TABSA, and MEABSA tasks.
- This paper considers the low-resource and polarity bias problems in the fine-grained sentiment analysis for the first time. Two data augmentation methods include entity replacement and noise injection are designed to deal with the problems.

## LITERATURE REVIEWS

### Research on fine-grained sentiment analysis

There are abundant researches on the ABSA task. LSTM (*Tang et al., 2016*) and an attention mechanism (*Wang et al., 2016*) have been applied to deal with the ABSA task in early time.

Following works include applying memory network-based (*Tang, Qin & Liu, 2016*) and attention-based (*Chen et al., 2017*) method to LSTM models, involving two stacked LSTMs (*Xu et al., 2020*) and so on. More recent models such as capsule network (*Chen & Qian, 2019*; *Du et al., 2019*), graph convolutional network model (*Zhang, Li & Song, 2019*), graph attention network (*Wang et al., 2020*), bi-level interactive graph convolution network (*Zhang & Qian, 2020*) are also used for ABSA task. *Zhu et al., (2019)* have exploited the interaction between the aspect category and the contents under the guidance of both sentiment polarity and predefined categories, and the proposed aspect aware learning framework has achieved satisfying performance in ABSA. The interactive relationships among aspect term extraction, opinion term extraction, and aspect-level sentiment classification have been investigated to encode collaborative signals for unified ABSA tasks (*Chen & Qian, 2020*). The pre-trained model such as RoBERTa has also been applied to improve ABSA with induced trees (*Dai et al., 2021*).

*Saeidi et al. (2016)* first proposed the TABSA task with SentiHood dataset. Following works include using additional commonsense knowledge (*Ma, Peng & Cambria, 2018*), developing a delayed memory update mechanism (*Liu, Cohn & Baldwin, 2018*), extending LSTM by adding the external knowledge (*Khine & Aung, 2019*) and so on. Additionally, Ye and Li proposed a recurrent entity memory network with word-level information and sentence-level hidden memory for TABSA (*Ye & Li, 2020*). In recent years, pre-trained language model is also applied to capture the dependence on both targets and aspects for sentiment prediction (*Wan et al., 2020*). BERT model has been applied to TABSA task. For example, auxiliary sentence has been found useful in TABSA when BERT model is applied (*Sun, Huang & Qiu, 2019*). Similarly, *Hong & Song (2020)* further fine-tune the pre-trained BERT model on SentiHood dataset. What's more, a context-guided softmax-attention and context-guided quasi-attention method is proposed to perform aspect categorization and TABSA at the same time (*Wu & Ong, 2020*).

*Yang et al. (2018)* first proposed the MEABSA task and contributed a dataset named *BabyCare*. They also proposed the Context memory, Entity memory and Aspect memory model (CEA) with RNN and deep memory networks. To improve the performance on long and complex text, an extended model of combining dependency trees with deep neural networks was proposed (*Yang et al., 2019*). The data sparsity challenge, also known as the cold-start problem, has also been investigated in MEABSA, which designed the frequency-guided attention mechanism to solve the problem (*Song et al., 2019*).

## Research on data augmentation in NLP

To alleviate the low-resource problem in various NLP tasks, data augmentations have been applied in previous works. The optional strategies mainly include word replacement, noise injection, text generation and so on. For example, it is useful to generate additional training examples that contain rare words in synthetically created contexts for machine translation (*Fadaee, Bisazza & Monz, 2017*). Another similar idea injected low-resource words into high-resource sentences to improve the low-resource translation task (*Xia et al., 2019*). Additionally, data augmentations such as synonym replacement and delexicalization have been applied to the NER task (*Dai & Adel, 2020*) and dialogue language understanding

(*Hou et al., 2018*) respectively. *Kim, Roh & Kim (2019)* proposed a method for spoken language understanding by introducing noise in all slots without classifying types of slots to improve the performance of low-resource dataset with "open-vocabulary" slots.

### Research on Bias problems in NLP

Bias, such as racial bias and gender bias (*Kiritchenko & Mohammad, 2018*; *Thelwall, 2018*), is also a trending topic of concern in different NLP researches. For example, *Zhao et al. (2018)* tried to mitigate gender bias by creating an augmented dataset identical to the original one by replacing the entities such as "he" or "she". Another work formally proposed the Counterfactual data augmentation (CDA) for gender bias mitigation in the coreference resolution task, by replacing every occurrence of a gendered word in the original corpus with its flipped one (*Lu et al., 2020*).

Recently, there are some related works to deal with the low-resource and polarity bias problems in coarse-grained sentiment analysis, which aims to predict the sentiments of the given posts. An early work introduced a bias-aware thresholding method motivated by cost-sensitive learning (*Iqbal, Karim & Kamiran, 2015*). Recent works include designing a sentiment bias processing strategy for the lexicon-based sentiment analysis (*Han et al., 2018*), and using the generation-based data augmentation method to deal with the low-resource problem in coarse-grained sentiment analysis (*Gupta, 2019*). To the best of our knowledge, there is no recent work discussing solutions to low-resource or polarity bias problems in fine-grained sentiment analysis.

## METHODS

ABSA, TABSA and MEABSA are three widely discussed tasks for fine-grained sentiment analysis, whose common objective is to predict the sentiment towards each aspect of each target entity. The detailed comparisons and examples can be found in Table 1 in the introduction section. This section introduces the methodologies, which we used to unify the ABSA, TABSA, and MEABSA tasks together with the same architecture. The proposed all-in-one solution to **P**redict sentiment towards **E**ntities and **A**spects is named **PEA**. Figure 1 demonstrates the graphical abstract of the PEA model.

Firstly, the unified problem setting of fine-grained sentiment analysis covering ABSA, TABSA and MEABSA is as follows.

### Problem Setting

Given a post $Post_m = [w_1, w_2, \ldots, w_T]$, with an entity set (if available) $E_m = entity_1, entity_2, \ldots, entity_{|E_m|}$ and an aspect set $A_m = aspect_1, aspect_2, \ldots, aspect_{|A_m|}$. For the words or multiple words in $Post_m$, which are corresponding to the entities or aspects in $E_m$ or $A_m$, we call them entity terms and aspect terms. The fine-grained sentiment analysis aims to predict the sentiment $y_{entity_i}^{aspect_j}$ towards the given $aspect_j$ of the certain $entity_i$ in $P_m$.

For the ABSA task, the entity set $E_m = \varnothing$ and the prediction target is simplified to $y^{aspect_j}$.

For the TABSA task, in each post $Post_m$, there is only one or two entities in the entity set, where $|E_m| = 1$ or $|E_m| = 2$. The prediction target becomes $y_{entity_i}^{aspect_j}$ towards all the aspects for the target entity in $Post_m$.

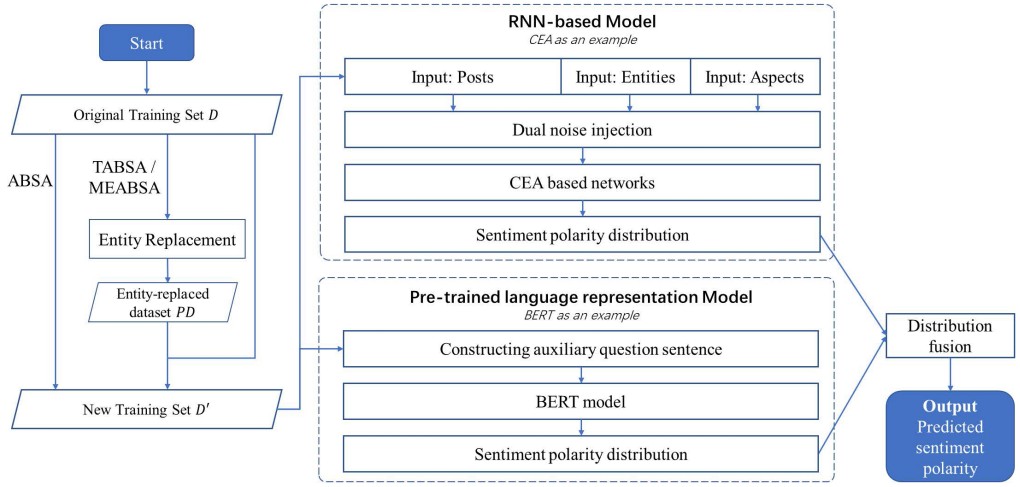

**Figure 1** The graphical abstract of the PEA model.

For MEABSA, the most challenging task, there are multiple entities and aspects in $Post_m$, where $|E_m| \geq 1$ and $|A_m| \geq 1$. It aims to predict $y_{entity_i}^{aspect_j}$ towards the mentioned aspects for every entity $entity_i$ in $Post_m$.

The general training workflow of PEA includes:

(1) Given an original training set $D$, generate a new training set $D'$ based on entity replacement. For the ABSA task, there is no entity involved, so the entity replacement step is skipped and $D' = D$. For TABSA and MEABSA, entity replacement is conducted to get an entity-replaced dataset $PD$, and $D' = D \cup PD$. The entity replacement used in PEA is introduced in the first part of subsection "Data Augmentation".

(2) An RNN-based model is trained on the new training set $D'$ as one of the basic models. The dual noise injection is conducted on the input posts, entities and aspects to get the noise-injected vectors. The dual injection used in PEA is introduced in the second part of subsection "Data Augmentation". Then, we take an attentional recurrent neural network-based model, CEA (*Yang et al., 2018*), as an example, to be the basic model, whose output is the predicted sentiment polarity distribution of the given inputs. It is introduced in the first part of subsection "Basic Models".

(3) A pre-trained language model is trained on the new training set $D'$ as the other basic model. Auxiliary question sentences are constructed for training the BERT-based model, which can predict fine-grained sentiment polarity distribution with the given inputs. The detailed design is described in the second part of subsection "Basic Models".

(4) Finally, the ensemble method is applied to fuse the predicted sentiment polarity distribution of the RNN-based and BERT-based model as the outputs of PEA, which is the final predicted sentiment polarity. The fusion strategy is introduced in the third part of subsection "Fusion Strategy".

## Data augmentation

Data augmentation is widely used to improve learning performance, prevent overfitting, and increase robustness under low-resource conditions. This section illustrates two innovative, task-specific data augmentation methods that are deployed in the model.

**Entity Replacement.** The low-resource problem in fine-grained sentiment analysis mainly comes from entities in the posts. This problem can be alleviated by increasing the low-resource entities. Among the data augmentation methods used in recent works for alleviating the low-resource problem in other NLP tasks, replacing words in context with similar ones is a viable data augmentation method (*Fadaee, Bisazza & Monz, 2017*; *Xia et al., 2019*; *Dai & Adel, 2020*). Usually, similar words can be extracted from word similarity calculation (*Wang & Yang, 2015*), and can also be extracted from a handcraft ontology such as WordNet.

In previous works, any word in a sentence can be replaced. This kind of replacement is extremely risky in fine-grained sentiment analysis tasks. For example, if a sentiment word, such as "happy", was replaced, it would unintentionally change the sentiment polarity at the same time. To avoid this kind of situation, we proposed the entity replacement method which successfully addresses this problem. Entity replacement is used to generate pseudo instances for training. The entire process involves 3 steps:

- Creating a duplicate of the original training set $D$.
- Replacing each entity in the duplicated dataset with the target entity to get an entity-replaced dataset $PD$.
- Combining the original dataset with the entity-replaced dataset as the new training dataset $D' = D \cup PD$ to train models.

In step 2, target entities are selected dynamically based on the scarcity of entities in the original training set so that every entity will have sufficient training instances eventually. In other words, the fewer times an entity presents in the original training set, the more likely it will be selected as the target entity. The detailed probability that an entity is selected is calculated as follows:

$$P(entity_i) = \frac{\left|mention(entity_i)\right|^{-1}}{\sum_{j=1}^{|E|}\left|mention(entity_j)\right|^{-1}}, \forall i \in [1,|E|] \tag{1}$$

where $E = \bigcup_{E_m \in D} E_m$ is the total entity set in the original training dataset $D$, $\left|mention(entity_i)\right|$ represents the number of instances mentioning $entity_i$ in $D.x^{-1}$ is an inverse proportional function, where $x^{-1} = \frac{1}{x}$.

Table 2 shows an example of such a replacement. Besides increasing the number of training instances, we think data augmentation also helps solve the polarity bias problem. For example, if an entity is always labeled positive in the training set, it will be more likely to be predicted positive no matter what the post is about. The proposed data augmentation methods help the polarity balance for entities because the entity may be replaced into a positive or neutral or negative expression randomly.

**Table 2  An example of entity replacement.** The replacement maintains the same sentiment polarity and correct grammar.

| | | |
|---|---|---|
| **Original** | **post** | I've used MacBookPro, it's convenient. |
| | **entity** | MacBookPro |
| **New** | **post** | I've used Thinkpad, it's convenient. |
| | **target entity** | Thinkpad |

To conclude, the low-resource entity replacement is designed to increase the number of training instances, especially for the low-resource entities, and help solve the polarity bias problem in sentiment prediction towards multiple entity settings.

**Dual Noise Injection.** To improve the generalization ability of PEA, we also involve the noise injection method. In previous NLP tasks, such as machine translation (*Cheng et al., 2018*) and spoken language understanding (*Kim, Roh & Kim, 2019*), it has shown the effectiveness of improving the model's generalization ability by injecting noises. In these works, noise is usually injected into the context representation for the post directly. For fine-grained sentiment analysis, the inputs include context texts, entities, entity terms, aspects and aspect terms. It is not applicable to only inject noises on context representations like previous works. Therefore, we propose the idea of dual noise injection: a noise is injected into the representation of entity and entity terms in the context at the same time. A similar practice is performed on the aspect and aspect terms.

In this task, the dual noise injection is used to simulate new entities and new aspects, enabling the model to make better predictions when it comes across low-resource entities or aspects. Following the common choice of previous works (*Cheng et al., 2018*; *Kim, Roh & Kim, 2019*), we also use the Gaussian noise to inject noises into the embedding space of posts, entities and aspects. Figure 2 is an example to illustrate the detailed processes of dual noise injection.

The dual noise injection consists of 3 steps:

- We first express the post, entity, and aspect in vectors space $v_w \in \mathbb{R}^{T \times k}, v_e \in \mathbb{R}^k, v_a \in \mathbb{R}^k$, where $v_w = v_{w_1}, \ldots, v_{w_T}$, $T$ represents the number of words in the post, and $k$ is the dimension of representations. The embedding vectors can be initialized by GloVe (*Pennington, Socher & Manning, 2014*).
- Then we sample noise vectors $n_e \in \mathbb{R}^k$ and $n_a \in \mathbb{R}^k$ for entity and aspect respectively from the Gaussian distribution.
- At last, we extract indicator vector $i_e = i_e^0, \ldots, i_e^T$ for entity terms indicating the location of entity terms in the post. Each element in $i_e$ is binary. $i_e^t$ is set to 1 when the $t^{th}$ word in the post is an entity term, otherwise, it is set to 0. Note that an entity term may consist of one or more words. In the same manner, we can get an indicator vector $i_a$ for aspect term. Then, we inject the noise to the entity, the aspect, and the post:

$$v_e' = v_e + n_e. \tag{2}$$

$$v_a' = v_a + n_a. \tag{3}$$
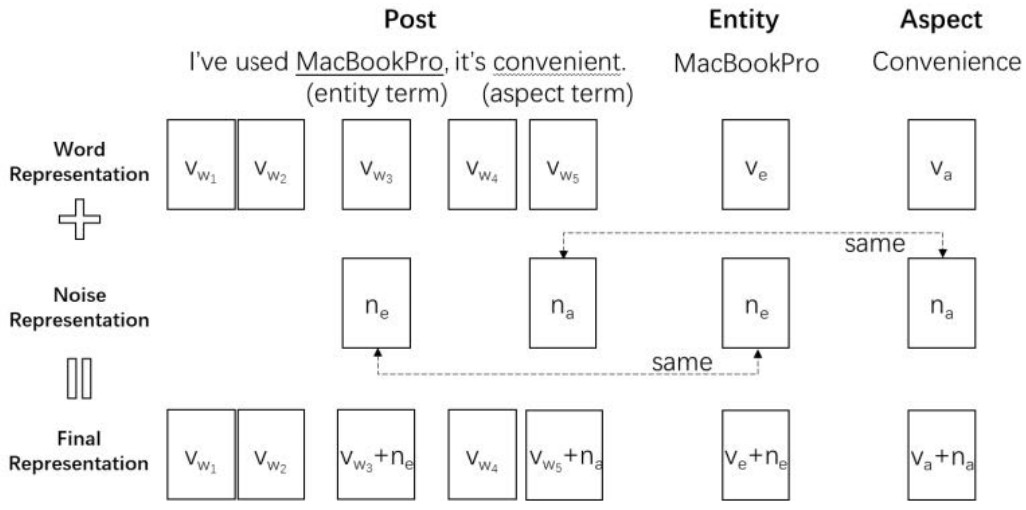

**Figure 2** **An example of dual noise injection.**

$$v'_{w_i} = v_{w_i} + i^e \times n_e + i^a \times n_a. \tag{4}$$

In step 2, the same noise vector (*e.g.*, $n_e$) needs to be applied to the entity and entity term. This is to ensure the new-generated entity and entity term remain the same relative location in the embedding space. We also apply the same noise vector (*e.g.*, $n_a$) to the aspect and the aspect term in the same manner. The noise injected into the entity and aspect does not have to be equal.

Also, if the noise level is not large enough, it won't substantially change the effect of injections. In order to test what is the best noise level in this case, we conduct experiments to determine the settings, which is introduced in section "Experimental Settings".

## Basic models

Recently, both RNN-based models and BERT-based models have shown effectiveness in the fine-grained sentiment analysis (*Yang et al., 2018*; *Yang et al., 2019*; *Sun, Huang & Qiu, 2019*; *Xu et al., 2020*). Due to the different structures of RNN and BERT, both kinds of models have advantages and weaknesses respectively. PEA incorporates both models to help make the final prediction more accurate.

## RNN-based model for fine-grained sentiment analysis

The CEA model is designed for MEABSA task, and can also be used for ABSA and TABSA tasks. It takes the word vectors of the post, the entity vectors and aspect vectors as inputs, and predicts the fine-grained sentiments towards the given aspect of the entity. To incorporate

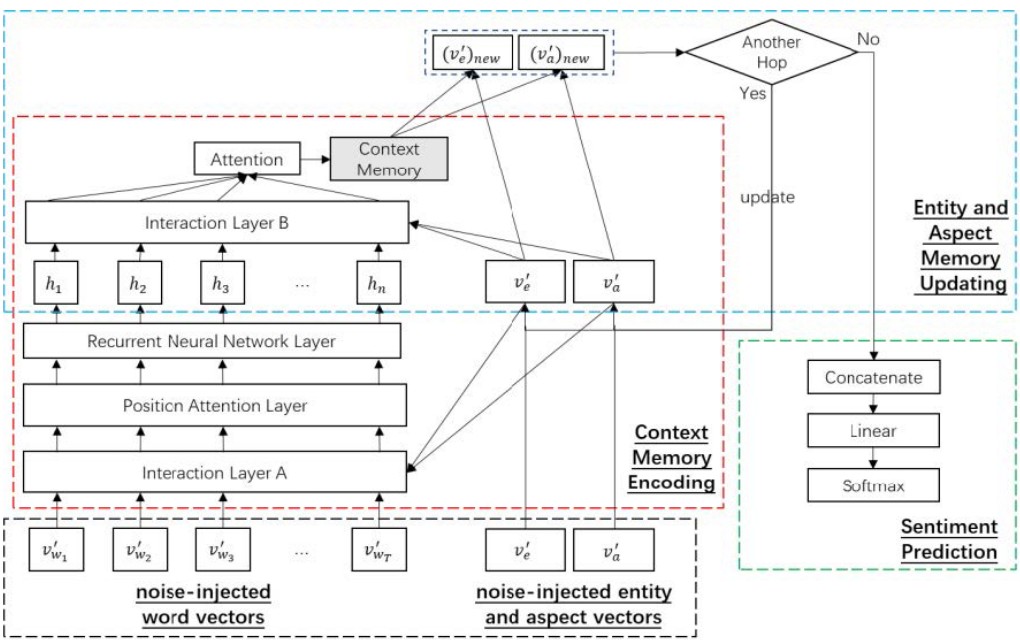

**Figure 3  General structure of CEA with noise-injected vectors.**

noise injection with CEA, we feed the noise-injected vectors to CEA, the general structure of noise-injected CEA is as Fig. 3 shows.

Firstly, we feed every noise-injected word vector $v'_{w_i}$ in the post to CEA. An LSTM layer is applied to extract the semantics of the post after a few data processing layers. After that, a deep memory network is applied to update entity and aspect representations with the given noise-injected entity vector $v'_e$ and aspect vector $v'_a$. The updated representations are fed into a dense layer to predict the final sentiment. For detailed explanation of CEA, refer to the original paper (*Yang et al., 2018*).

Because CEA requires entities and aspects as inputs, it is naturally suitable for the TABSA and MEABSA tasks. For the ABSA task, if there is no entity mentioned in the post, we can set the entity vector to a zero vector as the input. This makes the CEA-based basic model be able to deal with all the ABSA, TABSA and MEABSA tasks.

## Pre-trained language model for fine-grained sentiment analysis

The pre-trained language model is useful for enabling low-resource tasks to benefit from a huge amount of unlabeled data by pre-training. Bidirectional Encoder Representations from Transformers (BERT) (*Devlin et al., 2018*) is one of the key innovations in language representation learning (*Howard & Ruder, 2018*; *Peters et al., 2018*). It has achieved good results in many natural language processing tasks (*Acheampong, Nunoo-Mensah & Chen, 2021*; *Van Aken et al., 2019*).

BERT uses bidirectional pre-training for language representations, and it is pre-trained on two tasks: *masked language model* for understanding the relationship between words, and *next sentence prediction* for understanding the relationship between sentences for

downstream tasks. The design of pre-training makes use of a huge amount of unlabeled data, making it suitable for low-resource situations. Thus, we incorporate BERT to further enhance performance.

*Sun, Huang & Qiu (2019)* argued that constructing an auxiliary question sentence for the BERT model is useful in the TABSA task. We follow the conclusion and make the auxiliary question sentence for the entity and aspect with the template of "What is the sentiment towards the [aspect] of [entity]?". Then the sentiment classification task is turned into a sentence pair classification task. The label set of this setting includes {Positive, Neutral, Negative}. The BERT model takes two paragraphs as input with the token [CLS] at the beginning and [SEP] at the end of each paragraph. We set the post as the first paragraph and the auxiliary question sentence as the second. Here is an example.

| | |
|---|---|
| **Input:** | [CLS] I've used MacBookPro, it's convenient. [SEP] What is the sentiment towards the convenience of MacBookPro? [SEP] |
| **Output:** | Positive |

By constructing auxiliary question sentences along with the posts, we can generate inputs suitable for training BERT-based models, whose outputs are the predictions of sentiments towards targeted aspects of entities.

The construction of inputs can be applied to the TABSA and MEABSA directly. For the ABSA task, there is no entity mentioned in the post, the underlined part in the constructed question template, which is "What is the sentiment towards the [aspect] of [entity]?", will be omitted. This makes the BERT-based basic model be able to deal with all the ABSA, TABSA and MEABSA tasks.

## Fusion strategy

Ensemble methods can improve the predictive performance of a single model by training multiple models and combining their predictions. The weighting method is one of the effective strategies to fuse outputs, which assign weights to each basic model to combine the final decision (*Sagi & Rokach, 2018*), including simple averaging and weighted averaging (*Zhou, 2021*). We follow the strategy of simple averaging and combine the data augmented CEA with BERT to be the final model. We train the two models separately, and ensemble their predictions by taking the sentiment polarity with the largest averaged predicted probability as the final output. For a given post $Post_m$, the fine-grained sentiment prediction towards $aspect_j$ of $entity_i$, denoted as $y_{entity_i}^{aspect_j}$, is calculated as Eq. (5) shows.

$$P(c_i) = 0.5 \times P_{BERT}^{c_i}\left(Post_m, entity_i, aspect_j\right) + 0.5 \times P_{CEA}^{c_i}\left(Post_m, entity_i, aspect_j\right) \qquad (5)$$

$$y_{entity_i}^{aspect_j} = argmax\ P(c_i)$$

where $c_i \in \{positive, neutral, negative\}$, $P(c_i)$ represents the probability that the sentiment is $c_i$, $P_{model}^{c_i}\left(Post_m, entity_i, aspect_j\right)$ represents the predicted probability of the sentiment $c_i$ towards $aspect_j$ of $entity_i$ in $Post_m$ by the basic model BERT or data augmented CEA.

## Time complexity analysis

Compared with existing deep learning-based models, our proposed PEA model involves entity replacement, dual noise injection and prediction fusion as additional modules. The analysis of time complexity for these three parts is described as follows.

For entity replacement, we calculated the selected probability for every entity, whose time complexity is $O(E)$, where $E$ is the total number of entities in the dataset. We then traversed every instance and conduct entity replacement, whose time complexity is $O(N)$, where $N$ is the number of instances in the data set. The total time complexity of entity replacement is $O(E) + O(N)$.

For dual noise injection, we traversed every token in each instance to find the tokens referring to entity and aspect, whose time complexity is $O(T)$, where $T$ is the length of each instance. We added dual noises on all instances, whose time complexity is also $O(N)$. The total time complexity of dual noise injection is $O(T) \times O(N)$.

For prediction fusion, we fused the prediction with the weighted summation operation on every category for each instance, whose time complexity is $O(c) \times O(N)$, where $c$ is the number of categories of sentiments.

The total time complexity of extra operations in our proposed PEA model is $(O(E) + O(N)) + (O(T) \times O(N)) + (O(c) \times O(N))$.

## EXPERIMENTS AND ANALYSIS

In this section, we introduce the experimental settings and results to validate the effectiveness of our PEA model.

## Experimental settings

We evaluate four benchmark datasets of three tasks, including datasets in two languages: English and Chinese. Statistics of the used datasets are displayed in Table 3.

- Restaurant and Laptop are two datasets from SemEval 2014 (*Pontiki et al., 2014*) for ABSA. Both datasets are reviews in English and each review contains aspects and corresponding sentiment polarities, including positive, negative and neutral.
- SentiHood is a widely used dataset for TABSA (*Saeidi et al., 2016*). It consists of 5,215 sentences in English, and 3,862 of which contain a single aspect, the rest contains multiple aspects. Each sentence is annotated with a list of tuples, which are aspect, given entity and corresponding sentiment polarity, including positive and negative. The whole dataset is split into train, validation and test set.
- BabyCare is a large public dataset for MEABSA (*Yang et al., 2018*). It consists of babycare reviews in Chinese and each review is in the format of a list of tuples, which are context, aspects, corresponding entities and sentiment polarities, including positive, negative and neutral. The whole dataset is split into train, validation and test set.

### Common settings

For the BERT and CEA models, we use default parameters. For all English datasets, we use BERT-Base English models (https://github.com/google-research/bert) and 6B300d GloVe (*Pennington, Socher & Manning, 2014*) word embeddings (https://nlp.stanford.edu/projects/

**Table 3** Statistics of used datasets.

| Dataset | Language | Training set | Validation set | Test set | Task ABSA | TABSA | MEABSA |
|---------|----------|--------------|----------------|----------|-----------|-------|--------|
| Restaurant | English | 3,608 | – | 1,120 | ☞ | | |
| Laptop | English | 2,328 | – | 638 | ☞ | | |
| SentiHood | English | 3,650 | 522 | 1,043 | | ☞ | |
| BabyCare | Chinese | 29,354 | 3,682 | 3,677 | | | ☞ |

glove/). For the Chinese dataset, we use BERT-Base Chinese and the same word vectors provided by *Yang et al. (2018)*. For multi-word entity terms and aspect terms, we follow the preprocessing in previous works (*Yang et al., 2018*; *Song et al., 2019*; *Yang et al., 2019*). We use the average vectors of all the words in the entity/aspect term as the entity/aspect term vectors.

### Task specific settings

For ABSA task, the *Restaurant* and *Laptop* datasets are used for experiments. Because there is no entity in these datasets, so entity replacement in data augmentation is removed when implementing PEA. For TABSA task, the *SentiHood* dataset is used for experiments. Because aspect location is not given in this dataset, aspect noise injection is removed in this task. For MEABSA task, the *BabyCare* dataset is used for experiments. When implementing PEA, both entity replacement and noise injection are remained in this task.

### Data augmentation settings

We perform entity replacement on the training data for the whole dataset and merge the pseudo instances with original instances. According to the proposed entity replacement method, those entities, which are low-resource in the original training set, have a higher probability to be chosen for replacement. Table 4 lists the top 10 low-resource entities in the *BabyCare* dataset, and displays the number of instances that belong to every category for both the original training set and the entity-replaced dataset. We can observe that, for those low-resource entities, such as "Kabrita", the number of negative and neutral instances has significantly increased by using entity replacement. This can help relieve both the low-resource and polarity bias problems.

For noise injection, $\mu$ and $\sigma$ are two parameters to be determined. We follow the common setting in previous works (*Kim, Roh & Kim, 2019*) for $\mu$, which is $\mu = 0$. For $\sigma$, we conduct experiments on all four datasets with $\sigma$ ranging from 0.01 to 0.4 to quantify the noise level. Experimental results are in Fig. 4.

The $x$-axis refers to different values of $\sigma$, the $y$-axis refers to the Macro-F1 performance. Four lines with different kinds of marks refer to the results of four datasets. Experimental results show that when $\mu = 0$ and $\sigma = 0.05$, noise injection achieves the utmost performance on all tasks. We use this setting in the following experiments.

**Table 4** Top 10 low-resource entities in the BabyCare dataset, with the number of instances that belong to every polarity category for both the original training set and entity-replaced dataset.

| Entity | The original training set | | | The entity-replaced dataset | | |
|---|---|---|---|---|---|---|
| | Negative | Neutral | Positive | Negative | Neutral | Positive |
| 佳贝艾特 (Kabrita) | 0 | 0 | 26 | 69 | 243 | 361 |
| 可瑞康 (Karicare) | 0 | 0 | 17 | 108 | 382 | 527 |
| 君乐宝 (JunLeBao) | 0 | 0 | 15 | 121 | 409 | 571 |
| 咔哇熊 (Cowala) | 0 | 0 | 14 | 144 | 446 | 642 |
| 多美滋 (Dumex) | 1 | 0 | 64 | 22 | 84 | 219 |
| 太子乐 (Happy Prince) | 1 | 0 | 19 | 102 | 306 | 485 |
| 奶粉 (milk powder) | 0 | 0 | 19 | 102 | 304 | 479 |
| 欧贝嘉 (OuBecca) | 1 | 0 | 19 | 86 | 305 | 459 |
| 百立乐 (Natrapure) | 4 | 0 | 73 | 37 | 71 | 183 |
| 诺优能 (Nutrilon) | 2 | 0 | 42 | 44 | 146 | 227 |

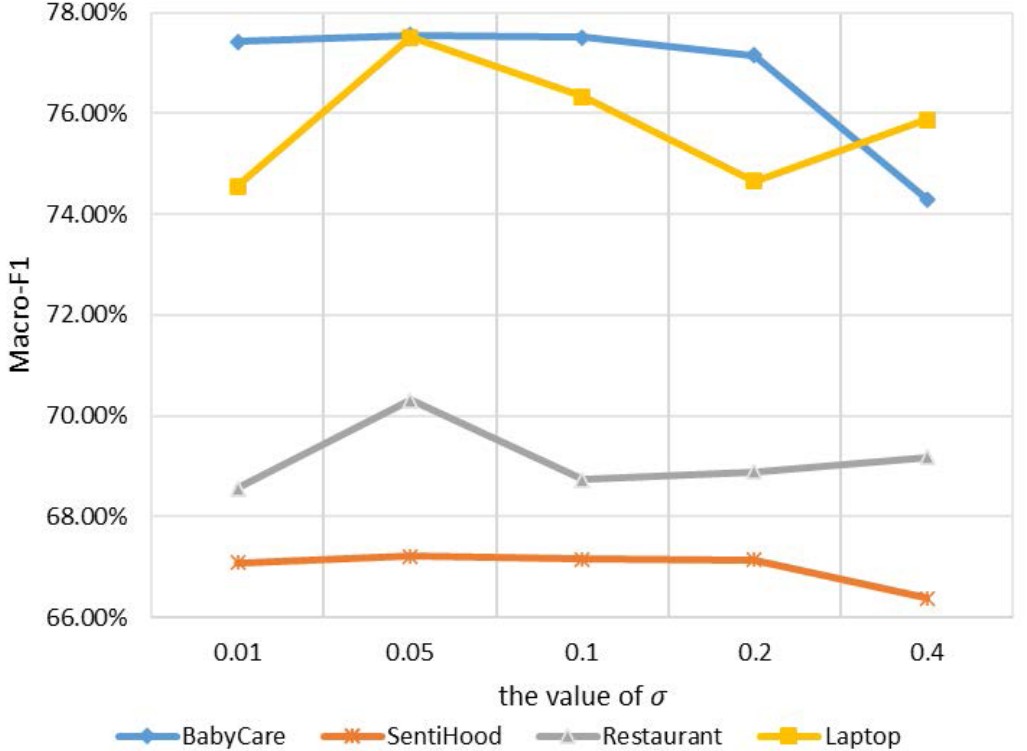

**Figure 4** Macro-F1 performance on four datasets with different values of $\sigma$ in noise infection.

### Model implementation settings

We implement our proposed model with TensorFlow 2.1, Python 3.7. The device we used consists of CPU (E5 2630 v4), GPU (1080ti * 4) and RAM (256G). We compare our model with the state-of-the-art baselines on 3 tasks predicting sentiment towards entities and aspects.

## Results

*Accuracy* and *Marco-F1* score are two main-stream metrics in most sentiment analysis research, where *Marco-F1* is the *F1* score averaged over all the classes. In the following experiments, *Marco-Precision, Macro-Recall and AUC score are also used according to different tasks.*

### Results on the ABSA Task

We evaluate the English benchmark datasets (http://alt.qcri.org/semeval2014/task4/) *Restaurant* and *Laptop* for the ABSA task. We compare with the published state-of-the-art baselines, including Target-Dependent Long Short-Term Memory (TD-LSTM) (*Tang et al., 2016*), MemNet (*Tang, Qin & Liu, 2016*), Attention-based LSTM with Aspect Embedding (ATAE-LSTM) (*Wang et al., 2016*), Interactive Attention Network (IAN) (*Ma et al., 2017*), Recurrent Attention on Memory (RAM) (*Chen et al., 2017*), Transfer Capsule Network (TransCap) (*Chen & Qian, 2019*), Aspect-specific Graph Convolutional Network (ASGCN) (*Zhang, Li & Song, 2019*), and Capsule Network with Interactive Attention (IACapsNet) (*Du et al., 2019*). Following the former research, *Accuracy* and *Marco-F1* are evaluated for both datasets, *Marco-Precision and Macro-Recall are also reported*. There is no entity in the dataset, so entity replacement in data augmentation is removed. Results on two ABSA datasets are shown in Table 5.

We can have the following observations:

(1) by observing the accuracy and F1 performance, two Capsule Network-based models TransCap and IACapsNet are much better than other previous baselines. This is because the key components of TransCap and IACapsNet are recurrent neural works and attention mechanisms. It shows that the RNN-based model has advantages in predicting fine-grained sentiments over conventional methods.

(2) by observing the precision and recall on both datasets, the recall scores of most models include TD-LSTM, ATAE-LSTM, IAN, RAM and ASGCN are much worse, while PEA can have better performance.

(3) compared with all the baselines, our proposed model PEA achieves significant improvements on both datasets. The experimental results show the PEA model is superior to other baselines in the ABSA task under all evaluation metrics.

### Results on the TABSA Task

We evaluate the English benchmark dataset *SentiHood* for the TABSA task. It consists of 5,215 sentences, 3,862 of them contain a single target, and the remainder multiple targets. We compare with all the published state-of-the-art baselines, including Logistic Regression (LR) (*Saeidi et al., 2016*), LSTM+TA+SA (*Ma, Peng & Cambria, 2018*), SenticLSTM (*Ma, Peng & Cambria, 2018*), Dmu-Entnet (*Liu, Cohn & Baldwin, 2018*), RE+Delayed-memory (*Liang et al., 2019*), BERT-pair-QA-B and BERT-pair-QA-M (*Sun, Huang & Qiu, 2019*). Following the former research in the TABSA task, *Accuracy* and *AUC* are usually reported and used as evaluation metrics, in the paper, *Marco-Precision, Macro-Recall and Marco-F1 are also reported*. Results on TABSA are presented in Table 6.

We can have the following observations:

**Table 5** Performance (%) on two datasets for the ABSA task, *Accuracy, Marco-Precision, Macro-Recall* and *Marco-F1* are reported.

| Models | Restaurant | | | | Laptop | | | |
|---|---|---|---|---|---|---|---|---|
| | Accuracy | Precision | Recall | F1 | Accuracy | Precision | Recall | F1 |
| TD-LSTM | 75.18 | 70.60 | 56.57 | 58.51 | 64.26 | 57.67 | 56.67 | 54.10 |
| MemNet | 77.32 | 69.87 | 64.38 | 64.61 | 68.65 | 63.58 | 63.62 | 62.69 |
| ATAE-LSTM | 74.38 | 67.43 | 57.28 | 58.32 | 66.14 | 61.22 | 58.97 | 56.91 |
| IAN | 76.16 | 67.43 | 59.31 | 60.56 | 65.20 | 61.64 | 58.54 | 54.08 |
| RAM | 76.07 | 72.07 | 58.65 | 59.59 | 68.03 | 64.03 | 63.86 | 60.82 |
| TransCap | 79.20 | 70.76 | 70.81 | 70.78 | 74.76 | 71.77 | 71.99 | 70.08 |
| ASGCN | 74.29 | 71.95 | 56.74 | 56.45 | 69.75 | 66.21 | 63.75 | 62.29 |
| IACapsNet | 81.79 | – | – | 73.40 | 76.80 | – | – | 73.29 |
| PEA(Our) | **84.82** | **80.41** | **76.31** | **78.14** | **78.68** | **74.43** | **76.60** | **75.07** |

**Table 6** Performance (%) on the SentiHood dataset for the TABSA task, *Accuracy, Marco-Precision, Macro-Recall, Marco-F1* and *AUC* are reported.

| Models | Accuracy | Precision | Recall | F1 | AUC |
|---|---|---|---|---|---|
| LR | 87.5 | – | – | – | 90.5 |
| LSTM+TA+SA | 86.8 | – | – | – | – |
| SenticLSTM | 89.3 | – | – | – | – |
| Dmu-Entnet | 90.2 | 74.8 | 76.3 | 75.5 | 94.8 |
| RE+Delayed-memory | 92.8 | – | – | – | 96.2 |
| BERT-pair-QA-B | 93.3 | – | – | – | 97.0 |
| BERT-pair-QA-M | 93.8 | 83.4 | **85.7** | 84.5 | 97.1 |
| PEA(Our) | **94.3** | **86.0** | 84.5 | **85.2** | **97.4** |

(1) BERT-pair-QA-M and BERT-pair-QA-B are the previous state-of-the-art models. Compared with other none-BERT based baselines, BERT-pair-QA-M and BERT-pair-QA-B outperform the LR, LSTM+TA+SA, SenticLSTM, Dmu-Entnet and RE+Delayed-memory models in both accuracy and AUC score. This result shows the effectiveness of the pre-trained language model for fine-grained sentiment analysis.

(2) compared with two BERT-based baselines, our proposed PEA achieves further improvement in most evaluation metrics. This may be because the prediction of PEA comes from both data augmented CEA and BERT, which helps ensemble the predictions of two basic models.

(3) different from the performance in ABSA and MEABSA, the improvement of PEA in the TABSA task seems slightly in accuracy and AUC score, this may be because aspect location is not given in this dataset (but given in other tasks), therefore, aspect noise injection is removed for this experiment. So we have conducted a statistical analysis test in the following section to show the performance difference between the two models is statistically significant.

### Results on the MEABSA Task

We evaluate the Chinese benchmark dataset *BabyCare* for the MEABSA task. We compare with all the published state-of-the-art baselines, including CEA (*Yang et al., 2018*), DT-CEA (*Yang et al., 2019*), Cold-start Aware Deep Memory Network (CADMN) (*Song et al., 2019*). These methods are exactly designed for this task. We also compare with MemNet (*Tang, Qin & Liu, 2016*), ATAE-LSTM (*Wang et al., 2016*), IAN (*Ma et al., 2017*), and their modified versions MemNet+, ATAE-LSTM+ and IAN+, which are used as baselines in a recent MEABSA work (*Song et al., 2019*). We follow the designs introduced in *Song et al. (2019)*: these three modified plus versions remain the basic model structure of MemNet, ATAE-LSTM and IAN respectively. The additional entities in the MEABSA task are treated as the aspects, and are added to the models in the same manner of aspects. These methods are originally designed for the ABSA task, and they are often regarded as baselines in former MEABSA research. Following the former research, *Accuracy* and *Marco-F1* are evaluation metrics for this dataset, *Marco-Precision and Macro-Recall are also reported*. Table 7 displays the comparisons between our model and baselines.

We can have the following observations:

(1) MemNet, ATAE-LSTM, and IAN in the first three lines only model aspects while ignoring entity modeling. Their performances are worse than the plus versions MemNet+, ATAE-LSTM+, and IAN+, which model the entity in the same manner as aspect, illustrating the effectiveness of entity modeling in the MEABSA task.

(2) The CEA model combines the advantages of both attention-based LSTM and deep memory networks, the former is the key component of ATAE-LSTM+ and the latter is the key component of MemNet+. The performance of CEA is much better than ATAE-LSTM+ and MemNet+, which reaches about 15% in accuracy. This shows that the CEA model has advantages in the MEABSA task, and is more suitable to be chosen as an RNN-based basic model for PEA.

(3) DT-CEA and CADMN are two extension models based on CEA. DT-CEA incorporated dependency information to improve CEA. CADMN used a frequency-guided attention mechanism to improve CEA. The performance of CADMN and DT-CEA are comparable to each other and are little better than CEA.

(4) compared with all the baselines, our proposed method PEA achieves significant improvement under all evaluation metrics. Compared with the previous state-of-the-art CADMN model, the improvements of PEA reach about 4% in accuracy and 5% in F1. The MEABSA is the most challenging fine-grained sentiment analysis task, this experimental result shows PEA has a significant advantage in the MEABSA task.

### Statistical analysis test

Refer to the previous works (*Li et al., 2020*), we conduct McNemars test as the statistical analysis test to further show the statistical difference between two models. *p*-value is the significance level, which means the performance difference between the two models. If the estimated *p*-value is lower than 0.05, the performance difference between the two models is statistically significant. Table 8 displays the *p*-values between PEA and other models on three sentiment analysis tasks respectively.

**Table 7** Performance (%) on the BabyCare dataset for the MEABSA task, *Accuracy, Marco-Precision, Macro-Recall* and *Marco-F1* are reported.

| Models | Accuracy | Precision | Recall | F1 |
|---|---|---|---|---|
| MemNet | 62.74 | 59.81 | 48.84 | 46.13 |
| ATAE-LSTM | 66.09 | 58.47 | 49.68 | 47.75 |
| IAN | 61.93 | 41.71 | 47.04 | 43.73 |
| MemNet+ | 65.32 | 59.93 | 50.55 | 47.93 |
| ATAE-LSTM+ | 66.25 | 56.01 | 51.93 | 51.87 |
| IAN+ | 65.81 | 44.42 | 50.06 | 46.50 |
| CEA | 80.20 | 77.68 | 75.23 | 76.29 |
| DT-CEA | 81.74 | – | – | 78.23 |
| CADMN | 81.45 | – | – | 78.37 |
| PEA(Our) | **85.72** | **83.97** | **82.60** | **83.25** |

**Table 8** *p*-value between PEA and other baselines on ABSA, TABSA and MEABSA tasks.

| | ABSA Task | |
|---|---|---|
| **Dataset** | **Restaurant** | **Laptop** |
| TD-LSTM | 1.4379e−14 | 1.4331e−14 |
| MemNet | 1.6116e−11 | 1.7323e−09 |
| ATAE-LSTM | 2.0494e−16 | 1.1331e−12 |
| IAN | 5.8819e−13 | 1.2102e−12 |
| RAM | 9.4895e−13 | 2.1595e−09 |
| TransCap | 1.1872e−06 | 0.0138 |
| ASGCN | 7.3462e−16 | 3.7338e−07 |

| | TABSA Task |
|---|---|
| **Dataset** | **SentiHood** |
| Dmu-Entnet | 6.7790e−41 |
| BERT-pair-NLI-M | 0.0174 |

| | MEABSA Task |
|---|---|
| **Dataset** | **BabyCare** |
| MemNet | 6.6475e−140 |
| ATAE-LSTM | 8.3216e−113 |
| IAN | 7.1802e−148 |
| MemNet+ | 3.4485e−120 |
| ATAE-LSTM+ | 1.1143e−114 |
| IAN+ | 1.6552e−114 |
| CEA | 2.2666e−20 |

We can observe that the performance differences between PEA and other baselines are statistically significant in all tasks, which show the effectiveness of the proposed PEA model from the perspective of statistical analysis. For example, in the TABSA task, the improvement of PEA compared with BERT-pair-NLI-M is not very high in accuracy,

which is 94.3% *vs* 93.8% in Table 6. In the statistical analysis test, the estimated *p*-value between PEA and BERT-pair-NLI-M is 0.0174. According to the definition of *p*-value, it shows that the performance difference between BERT-pair-NLI-M and PEA is statistically significant. Additionally, by observing Table 7 and Table 8 together, we can find PEA has significant advantages in the most challenging MEABSA task.

## ABLATION STUDY

Experimental results so far show that the PEA approach is superior to the baselines on all the ABSA, TABSA and MEABSA on selected datasets. Because PEA consists of data augmented CEA and BERT, we would like to further investigate the effectiveness of each part in the model. A case study is also introduced in this section.

### Effectiveness of components in PEA

Ablation study is used to show how each part of the model affects the performance by removing them. We conduct experiments on all four datasets of three tasks for comparisons. Experimental results are as Table 9 shows.

The proposed PEA model integrates data augmented CEA and BERT. Because entity replacement and noise injection are applied to data augmented CEA, we use CEA, CEA+EntityReplacement (CEA+ER for short) and CEA+EntityReplacement+NoiseInjection (CEA+ER+NI for short) respectively for ablation study to show the effectiveness of applying two data augmentation techniques. The BERT-based model is also used for comparisons in ablation studies.

We can have the following observations from Table 9:

(1) comparing CEA and CEA+ER, we can find involving entity replacement can have improvement on MEABSA and TABSA tasks. We also counted the number of instances for every entity based on the original training set and the entity-replaced dataset. The statistics are demonstrated with the box plot in Fig. 5.

It shows that using the proposed entity-replacement method can significantly increase the number of instances for low-resource entities, and all entities have at least 252 instances for training. For ABSA, there is no entity provided in the dataset, so the entity replacement procedure is removed.

(2) by adding noise injection, the CEA+ER+NI model achieves about 1.3% improvement on the Restaurant dataset over the CEA+ER model, and achieves slight improvement on other datasets. These observations show that using entity replacement and noise injection can bring positive impacts on fine-grained sentiment analysis. This may be because using data augmentation can increase the number of training instances, especially for low-resource entities and aspects, and help overcome polarity bias.

(3) by comparing the performance of PEA with the BERT-based model and data augmented CEA model, PEA achieves the best performance in most cases. The strength of BERT-based model is that it makes use of a huge amount of unlabeled data by pre-training, but it also has weaknesses. The BERT model depends on the Transformer (*Vaswani et al., 2017*), which further mainly relies on its self-attention mechanism. It has been suggested that self-attention has limitations that it cannot process input sequentially (*Dehghani et al.,*
**Table 9 Performance (%) of ablation study on four datasets.**

| Ablation | BabyCare | | SentiHood | | Restaurant | | Laptop | |
|---|---|---|---|---|---|---|---|---|
| | Accuracy | F1 | Accuracy | AUC | Accuracy | F1 | Accuracy | F1 |
| CEA | 80.20 | 76.29 | 90.3 | 93.2 | 78.13 | 67.96 | 71.41 | 67.05 |
| CEA+ER | 80.78 | 77.33 | 91.1 | 93.3 | – | – | – | – |
| CEA+ER+NI | 81.06 | 77.55 | 91.3 | 94.0 | 79.45 | 70.31 | 71.83 | 67.22 |
| BERT-based model | 84.12 | 81.62 | 93.8 | 97.1 | 83.52 | 76.11 | 76.99 | 72.40 |
| PEA (our) | **85.72** | **83.97** | **94.3** | **97.4** | **84.82** | **78.14** | **78.68** | **75.07** |

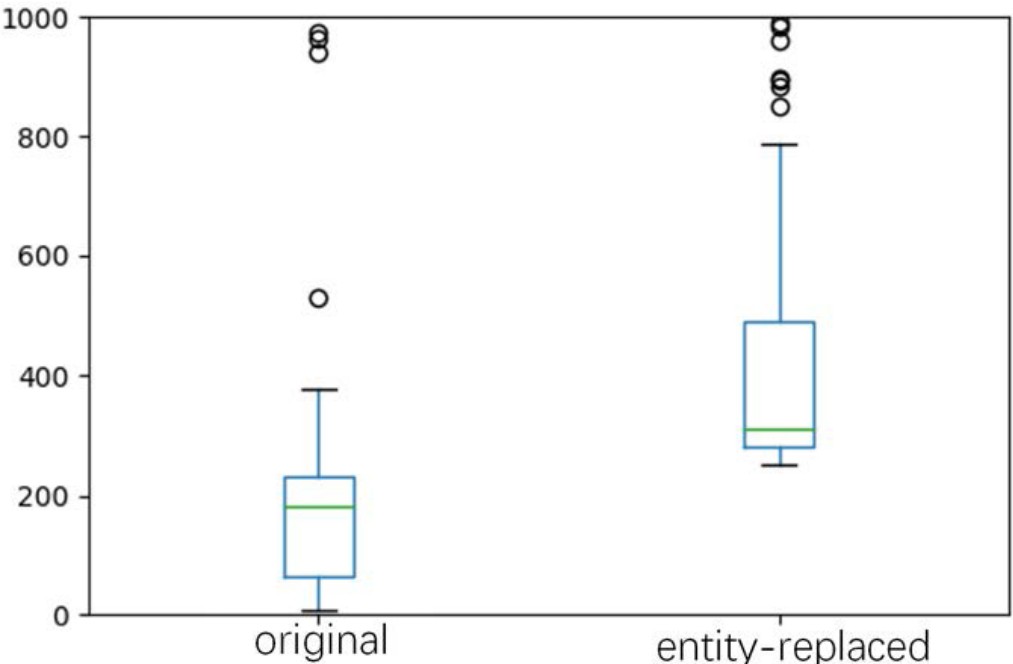

**Figure 5 Box plot of the number of instances for every entity based on the original training and the entity-replaced dataset, respectively.**

*2018*; *Hao et al., 2019*; *Shen et al., 2018*; *Hahn, 2020*). Such a weakness is just the strength of recurrent neural networks, which is one of the core components in CEA. Our model PEA combines the advantages of both and performs the best in most cases. To better understand the strengths and weaknesses of data augmented CEA and BERT, we carry out a case study in the next section.

## Case study

We give empirical validation on the strengths and weaknesses of two basic models, including the BERT-based model and data augmented CEA, by a further case study on misclassifications of both models. We test on the most challenging task MEABSA and use the corresponding *Babycare* dataset for the case study. To show the stability of the models rather than the occasionality, we have trained the BERT-based model, the Data augmented

**Table 10    Case study on misclassifications of BERT-based and data augmented CEA model.** The straight underlined words are entity terms and the wavy underlined words are aspect terms.

| | Example 1 | Example 2 | Example 3 |
|---|---|---|---|
| **Input** | I tried Pampers and Inherent. I dislike the smell of the former, and the latter leaks. | QiAn feels thin. I have brought BobDog, it will be delivered home. | I am too poor to afford Kao. My son and daughter are using XWW, it is cheap. |
| **Entity** | Pampers | BobDog | Kao |
| **Aspect** | Anti-leakage | Thickness | Cost |
| **BERT-based model output** | Negative | Neutral | Neutral |
| **Data augmented CEA output** | Neutral | Positive | Neutral |
| **PEA output** | Neutral | Neutral | Neutral |
| **Gold output** | Neutral | Neutral | Negative |

CEA model and the PEA model five times. The predictions of two representative examples are as Table 10 shows.

For example 1, the BERT-based model makes the same misclassification on the inputs five times and the data augmented CEA model achieves the correct predictions. Example 2 is just the opposite. Such stable misclassifications reveal the defects of both models.

The first example has a special pattern: the coreference structure of "...the former...,...,the latter...". The second example consists of two simple sentences. Correctly predicting the first example need the ability of global sequence or structure understanding which is the advantage of recurrent neural networks. The recurrent neural network is one of the core components of CEA. Correctly predicting the second example need the ability of local attention which is the advantage of self-attention, which is the core component of the BERT-based model. PEA fuses the prediction with both BERT-based model and data augmented CEA model based on ensemble methods, which make the correct prediction on both examples. This case study further helps illustrate the value and necessity of ensembling two basic models.

We also give the third example in Table 10, where all the CEA, BERT-based model and PEA made the wrong prediction. The gold output should be negative, but all models predicted it as neutral. The possible reason is that there are no aspect terms directly towards the target entity 'Kao', which cause the model to give the prediction as neutral.

# RESULTS ON CHALLENGING CONDITIONS

There are two challenges in sentiment prediction towards entities and aspects: the low-resource problem and the polarity bias problem. In this section, we evaluate the negative effect of challenges and the ability of models to solve them.

## Results on extreme low-resource conditions

To further test the model's performance under extreme low-resource conditions, we randomly selected 5%, 10%, 20%, and 50%, each time, from the original dataset as our

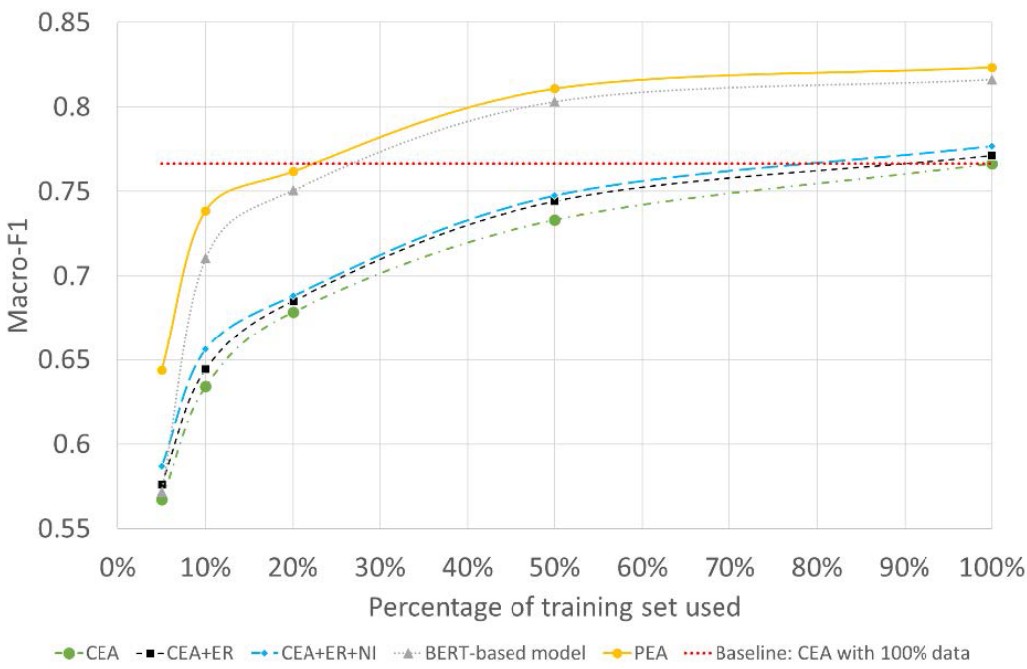

**Figure 6** **Performance on extreme low-resource conditions.**

training dataset. All tests are performed under the most challenging *Babycare* dataset. Experimental results are as Fig. 6 shows.

The *x*-axis refers to the percentage of data used for training, the *y*-axis refers to the Macro-F1 of different models. ER and NI are the abbreviations of entity replacement and noise injection. We can have the following observations from Fig. 6. (1) for all models, as the percentage of the training set used decreases, the models' performance drops significantly, which further illustrates the significance of the low-resource problem on sentiment prediction. (2) CEA+ER outperforms the CEA model under all the low-resource conditions, which shows the effectiveness of using entity replacement. By using noise injection, the CEA+ER+NI achieves further improvements over CEA and CEA+ER. (3) for the BERT-based model, when the resource is extremely low, the BERT-based model deteriorates sharply. For example, when 5% of data is used for training, the Macro-F1 of BERT-based model and PEA is 57.16% *vs* 64.37%. This shows that the combination of the data augmented CEA and BERT-based model for PEA can boost the stability of the model. (4) the dotted line in red refers to the baseline results with 100% data for training, we can observe from Fig. 6 that when only 20% data are used for training, the proposed PEA can achieve a similar performance of the CEA model with full-resource data for training. With the size of training data becoming larger, the improvement of PEA becomes more obvious. This shows the PEA model, which combines data augmented CEA with BERT-based model, has advantages under low-resource conditions.

**Table 11** *Macro-F1* **and standard deviation of** *Macro-F1* **(in the brackets) on evident polarity biased (EPB) test set and original test set.** "DA" is short for Data Augmentation.

| Models | EPB test set | Original test set | Decline on EPB |
|---|---|---|---|
| CEA | 0.7542 (0.0123) | 0.7714 (0.0040) | 1.72% |
| CEA+DA | 0.7753 (0.0069) | 0.7768 (0.0036) | 0.15% |
| BERT-based model | 0.8068 (0.0070) | 0.8162 (0.0040) | 0.94% |
| PEA | 0.8153 (0.0069) | 0.8234 (0.0061) | 0.81% |

## Results on evident polarity biased conditions

Polarity bias occurs when sentiment polarity distribution towards an entity is not uniform. Polarity bias reduces the performance when sentiments towards an entity diverge in the training set and in the test set (*e.g.*, 70% of sentiment towards entity A are positive in the training set while 60% of which are negative in the test set). We create a new test set named EPB test set, which consists of all the instances with entities polarity biased from the original test set. Using the *BabyCare* test set, we find entities in 30% of data (1,070 out of 3,677) have the evident polarity bias problem. Experimental results are as Table 11 shows, the last column displays the decline between the performance on the Original test set and EPB test set.

After comparing the sentiment prediction results from using the evident polarity biased data with the results from using the origin data, we have the following observations:
(1) the performance of all models has varying degrees of decline on the polarity biased EPB dataset. This shows the polarity bias problem is one of the challenges in fine-grained sentiment analysis.
(2) comparing CEA and CEA+DA, the performance on the EPB test dataset is close to the performance on the original test set. This is because data augmentations can relieve the polarity bias problem by providing plenty, omni-polar sentiment training data, and reduced the variance of test results to offer more stable performance. This shows applying data augmentations can address the polarity bias problem in fine-grained sentiment analysis and make the model more generality.
(3) comparing CEA and the BERT-based model, the performance on the original test set of the BERT-based model has a significant improvement than that of CEA.
(4) PEA achieves the best performance on the original test set, and relieves the polarity problem on the EPB test at the same time, which also shows the necessity and effectiveness of using the ensemble methods to fuse the predictions of CEA and BERT based models with data augmentations.

## CONCLUSIONS

In this paper, we developed the PEA model, which unified the ABSA, TABSA, and MEABSA tasks together for the first time and provided an all-in-one solution to interpret consumers' opinions on all kinds of social media platforms. For the first time, we analysed the effect of the sentiment polarity bias problem in these tasks. Most importantly, we created two innovative, task-specific methods to alleviate the low-resource problem and the polarity bias

problem, not only getting promising experimental results, but also providing inspiration for successors to make more contributions in this area. For future work, there are two possible extensions worth considering. The first one is to look for new ways to combine pre-trained language models with RNN-based models, to integrate both advantages. The second one is to further investigate more types of fine-grained sentiment analysis, and propose unified models handling various fine-grained sentiment-related tasks, for example, emotion cause analysis.

## ACKNOWLEDGEMENTS

We would like to sincerely thank our colleagues, Professor Xiao-jun Wu, and Jun Sun from Jiangnan University, Professor Jun-yuan Xie, and Chong-jun Wang from Nanjing University, for their kind support and guidance.

### Funding

This research was funded by the National Key Research and Development Program of China (No. 2020YFA0908300), the National Natural Science Foundation of China (Grant No.62002137, No.62006097), the Fundamental Research Funds for the Central Universities (No. JUSRP12021), in part by the Natural Science Foundation of Jiangsu Province (Grant No. BK20200593) and the State Key Lab. for Novel Software Technology, Nanjing University, P.R. China (No. KFKT2020B02). The funders had no role in study design, data collection and analysis, decision to publish, or preparation of the manuscript.

### Grant Disclosures

The following grant information was disclosed by the authors:
The National Key Research and Development Program of China: 2020YFA0908300.
National Natural Science Foundation of China: 62002137, 62006097.
The Fundamental Research Funds for the Central Universities: JUSRP12021.
The Natural Science Foundation of Jiangsu Province: BK20200593.
The State Key Lab. for Novel Software Technology, Nanjing University, P.R. China: KFKT2020B02.

### Competing Interests

Jun Yang is employed by Marcpoint Co., Ltd. Heng-yang Lu, Cong Hu and Wei Fang declare that they have no competing interests.

### Author Contributions

- Heng-yang Lu conceived and designed the experiments, performed the experiments, analyzed the data, prepared figures and/or tables, authored or reviewed drafts of the paper, and approved the final draft.
- Jun Yang conceived and designed the experiments, performed the experiments, analyzed the data, performed the computation work, prepared figures and/or tables, authored or reviewed drafts of the paper, and approved the final draft.

- Cong Hu and Wei Fang conceived and designed the experiments, authored or reviewed drafts of the paper, and approved the final draft.

## Data Availability

The datasets for ABSA (Restaurant and Laptop) are available at SemEval-2014: https://alt.qcri.org/semeval2014/task4/index.php?id=data-and-tools

The dataset for TABSA (SentiHood) is available at GitHub: https://github.com/uclnlp/jack/tree/master/data/sentihood

The dataset for MEABSA (BabyCare) is available at GitHub: https://github.com/jncsnlp/MEABSA.

## Supplemental Information

Supplemental information for this article can be found online at http://dx.doi.org/10.7717/peerj-cs.816#supplemental-information.

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
