# Peer review of "One for “All”: a unified model for fine-grained sentiment analysis under three tasks"

_PeerJ Computer Science, doi:10.7717/peerj-cs.816_

## Round 0.1 · original submission · Major Revisions

Detailed reports have been received. I agree with the comments. They should be very helpful for improving the paper. Please provide a detailed response letter.

Reviewer 1 ·

Basic reporting

- The writing of he paper can be improved across the entire paper. I am providing suggestions and line numbers for the authors to make some of the corrections.
- The main issue of the paper which made it very hard for me to read is that it is very wordy and the tables and the figures are only appearing at the end. So it's very hard to read the text on results without looking at numbers and figures. I strongly recommend that authors address this.
- Authors can also break text into paragraph in places where a paragraph is too long and provide a title for the paragraph to separate different aspects or the parts of the discussion. This is specifically needed when reporting results.
- I find the case study section very difficult to read and it also focuses on one or two examples. It does not really provide much insight. Also, you have only included examples where the PEA model does well. It is much more helpful to know which examples it can not get right. First of all, if you bring the relevant table to be next to the section, it can make it easier to understand the text. But in general, if you want to gain extra space, I would remove this section.
- When reading through the paper, it's not clear from the method section that PEA will be a unified approach for all the tasks. Bringing Figure 1 to the beginning of the paper will help clarify this.
- There are some exceptions for different tasks and datasets but these are only mentioned when reporting the results. These should be in the experimental set up at least.


Detailed Comments:
- in all the paper ‘pre-trained language representation model’ can be replaced with ‘pre-trained language model’
- Line 115: it’s strange “which makes both advantages of RNN-based models and
BERT-based models with ensemble methods”
- Lines: 128-131 can be rephrased: graph neural networks (ref1, ref2) have been applied to the problem. - Also, stacked LSTM can be mentioned closer to the LSTM related work.
- Related work: the term ‘associates’ has been used a lot, it’s best to rephrase some sentences to avoid this.
- If the formatting allows, it’s a good idea to have section or paragraph titles in the related work to separate different aspects.

- In the methods section, PEA is the name of the entire solution including data augmentation. It’s better to do a separation: Data Augmentation, Baseline Models, etc.
- In line 215: the fusion strategy should not be part of the basic models and part of the proposed model.
- 231: we invented is a strange word, maybe use propose.
- Equation (1), |entity_i| is a misleading notation. Maybe we can say: |mention(entity_i)|
- 247: Table 2 shows an example …
- Lines 260-262, the sentence starting with ‘However’, I don’t understand what we mean by it. Can you please rephrase
- 272: ‘T is the length of the post, which is the number of words in the post’, the two are redundant, use one.
- 290: we conduct experimental attempts to determine the settings
- 296: basic models: basic can be dropped
- 307: ‘Detailed explanations of CEA can refer to the original paper’: for detailed explanation of CEA, refer to the original paper
- 311: Here, you mention that you deal with all the three tasks using the same architecture. You need to make it more clear early on.
- 317: It achieved good results in .. -> I has achieved good results (you should add references on tasks where BERT has achieved good results)
- 347: sentimental should be just sentiment
-390: ‘experimental attempts’ -> experiments
- 401: ‘researches’ -> research (replace in all the paper)
- 431: ‘while PEA can have stable performance’ what do we mean by stable performance?
- 417: you need to have a new section called Results. All the paragraph names should have ‘Results’ instead of ‘Evaluation’
- 400-416: In my opinion, the definition of the metrics (equations) can be removed. But do explain what macro F1 is (F1 averaged over all the classes).
- 447: This result shows that the effectiveness -> This result shows the effectiveness
- 450: ‘This may be due to the prediction of PEA comes from’-> This may be because the prediction …..
- 452: what is the point of (3), it’s not clear. Do you mean that the increase in performance is not very high?
- 454: ‘aspect noise injection is removed for this experiment’ : this is the first time this is mentioned. It should have been mentioned in the experimental set up.
- 488: the definition of p-value can be improved. What does ‘The -value is set as 0.05’ means, I think that can be removed.
- Lines 495-499: very confusing as to which ones are significant and which one is not significant. Which models does this sentence refer to ‘The estimated -value between these two models is 0.0174, which is lower than 0.05’?
- 502-503: ‘Experimental results in section “Experiments and Analysis” illustrated the PEA model’s outstanding performance in ABSA, TABSA and MEABSA than the baselines.’ -> experimental results so far show that the PEA approach is superior to the baselines on all the ABSA, TABSA and MEABSA on selected datasets.
- 559: ‘It is introduced that there are two challenges’ -> ‘There are two challenges’
- 575: instead of ‘seriously’, use another word, maybe sharply or massively.
- 576: ‘​​This shows that combine the’ -> This shows that the combination of ..
- 611: ‘Moreover, we first-time discovered and defined’ -> For the first time, we analysed the effect of the sentiment polarity bias’
- Table 4 is not necessary, you can remove it.

Experimental design

Experimental design seems fine. One comment is that how did the authors came up with the weights 0.5 for the ensemble model? Is it possible to optimize this hyper-parameter?

Validity of the findings

The findings seem valid and they will be easier to understand if the reporting improves.

Reviewer 2 ·

Basic reporting

no comment

Experimental design

Entity replacement is good for low-resource problem, it's also good for the polarity bias problem. It would be better to give some examples of entity replacement results on polarity biased entities.

Validity of the findings

I'm interested in the application of this model. If we want to apply the model to real scenarios, what should the data be like to input to the model.

Additional comments

1. The low-resource and polarity bias problems are indeed two challenges in the research field of fine-grained sentiment analysis. The experiments show good results.
2. The design of "dual noise injection" is interesting, it models even non-existed entities and aspects. This improves the generalization ability of the model and the experiments verified it.
3. The case study setting is good for understanding the strength and weakness of RNN-based model and BERT-based model, and explains why the two models should be combined.

Reviewer 3 ·

Basic reporting

1. Expansion of abbreviations like CEA, RNN, TD-LSTM, ATAE-LSTM should be given the first time the abbreviation is used
2. There are several works on Aspect-Based Sentiment Analysis, but very few on Targeted Aspect-Based Sentiment Analysis and Multi-Entity Aspect-Based Sentiment Analysis. So, literature review on TABSA and MEABSA could be elaborated in detail. Even ABSA literature work could be detailed
3. Racial and gender bias is discussed in literature review, is it implemented in the proposed work?

Experimental design

1. The various attributes of the datasets can be explained, so as to keep the reader engaged.

Validity of the findings

1. According to this work, what is the scope of fine – grained sentiment analysis? How do you differentiate it with coarsely grained sentiment analysis? Explain it beforehand
2. Figure is cited in text, which makes few things unexplained. For example, Figure 1 gives a prediction as output, What is predicted here?

Additional comments

1. Few more examples can be added to explain ABSA, TABSA and MEABSA for better clarity.

Annotated reviews are not available for download in order to protect the identity of reviewers who chose to remain anonymous.

---

## Round 0.2 · accepted · Accept

The paper can be accepted. Congratulations!

Reviewer 3 ·

Basic reporting

no comment

Experimental design

no comment

Validity of the findings

no comment

Additional comments

no comment